# Impacts of social isolation and risk perception on social networking intensity among university students during covid-19

Hyeon Jo[1], Eun-Mi Baek[2]*

1 Department of Planning, RealSecu, Busan, South Korea, 2 Department of Preventive Medicine, College of Medicine, Catholic University of Korea, Seoul, South Korea

* hanel2004@naver.com

## Abstract

This paper aims to examine the impacts of social isolation and risk perception on social networking intensity during COVID-19. Data was gathered from 345 university students. The present study empirically analyzed the data through a partial least squares methodology. The analysis showed that perceived behavioral control positively impacts social networking intensity. Affective risk perception positively affects subjective norms and perceived behavioral control. Cognitive risk perception has a significant association with both subjective norms and perceived behavioral control. Moreover, cabin fever syndrome serves as the key determinant of both sub-scales of risk perception. This study is novel in that it organically examines the effects of risk perception, social action, and closure on social networking. The current research and findings will offer useful implications for service providers in the social network industry.

## 1. Introduction

The world is undergoing various social, economic, and cultural changes due to COVID-19 [1]. People are restricted from outside activities, wear masks, and enter their information when visiting restaurants or stores [2]. University students have taken online lectures during the COVID-19 crisis and campus activities almost disappeared [3, 4]. The students supplement reduced external activities and social exchanges by performing social networking in cyberspace. They conduct social networking to check the daily life of others, communicate with friends, and express their thoughts. Sometimes, students use social network apps [5] or social media [6, 7] to enhance academical performance. Through social networking at home, students can relieve psychological frustration while hindering the dissemination of COVID-19. In this sense, it is valuable to explain the social networking behavior of university students during the pandemic. Thus, this paper identifies the drivers that influence the changed social networking use by focusing on disaster situations.

People see COVID-19 every day through news or online media and recognize the seriousness of the situation. Citizens may be emotionally aware of the danger by hearing from people around them about the confirmed cases or deaths resulting from COVID-19. Risks can also be

**Data Availability Statement:** All relevant data are within the manuscript and its Supporting Information files.

**Funding:** The author(s) received no specific funding for this work.

**Competing interests:** The authors have declared that no competing interests exist.

recognized numerically through government statistics reported on TV or news. University students who feel a stronger sense of risk will form a greater degree of subjective norm and perceived behavioral control over social measures to avoid risk and pursue personal safety. Risk perception was verified as a major antecedent variable determining subjective norms and perceived behavioral control for social distancing [8].

People experience emotional discomfort while being trapped in one place for a long period, which is called cabin fever [9]. Social measures against COVID-19 cause physical isolation, which in turn leads to individual psychological exhaustion [10]. Several works have proposed many solutions to cope with cabin fever syndrome [11–13]. Solutions include interacting with others in a virtual space. [9, 14]. Based on this, university students may further increase social networking activities to relieve their frustration and helplessness. The students with stronger cabin fever syndrome suffer from both the dangers of COVID-19 itself and the emotional discomfort caused by isolation measures to evade prevention.

University students hear from their friends and family about the COVID-19 situation. They become to know information such as the level of social distancing, period of social measure, the number of confirmed cases, and cases in other countries. Among them, people's views or positions on social measures form individual subjective norms. Subjective norms significantly lead to behavioral intention [15, 16]. Perceived behavioral control over social measures also influences individuals' behavior under COVID-19 [8, 17]. Subjective norms and perceived behavioral control may impact the overall behavior intention of university students in the COVID-19 environment, which will also be applied to social networking activities. This is because students with stronger level of subjective norms and perceived behavioral control on social measure have a tendency to spend more time at home. They also would supplement their reduced social activities with social networking on the internet space.

To sum up, COVID-19 put people at risk. Society has implemented various measures to respond to it. These measures restricted people's activities and made them feel socially isolated. They would be more involved in social measures to alleviate the risk and isolation. People will increase online social networking if they are more encouraged to participate in social measures and have more capacity to follow measures. This is because social networking compensates for reduced physical networking with activities that can be done at home. In addition, they may become more active in online social networking to relieve risk and feelings of isolation.

Risk perception, social measures (e.g. social distancing), and feelings of closure have a causal relationship. The spread of COVID-19 has made people aware of risk. To prevent infection, humans have implemented various social measures. Social measures have increased feelings of claustrophobia by curbing outdoor activities and increasing time spent at home. Risk perception, social measures, and feelings of closure may organically influence human behaviors. However, the problem with existing studies is that they introduce these three variables independently [18, 19] or selectively include two measures [20, 21]. This study is of academic significance because it addresses this research gap. In this sense, we posit the following research questions.

1. Do people with higher levels of risk perception seek to comply with social measures more?

2. Do people who feel more enclosed try to comply with social measures more?

3. Does closedness affect social networking behavior directly?

The objectives of this study are to 1) examine how risk of COVID-19 and personal dispositions toward lockdown influence responses to social measures, 2) identify the impact of individual responses to social measures on social networking activity, and 3) test the direct impact of feelings of lockdown on social networking.

The paper is organized as follows. Section 2 lists prior research related to social networking, risk perception, and cabin fever syndrome. Section 3 describes the research model and hypotheses. Section 4 presents the data collection and measurement tools for the empirical analysis. Section 5 presents the results of the study. Section 6 contains a discussion of the results. Section 7 presents theoretical and practical implications. Finally, Section 8 discusses the limitations of the study and future research directions.

## 2. Related work

### 2.1. Social networking

With the rapid spread of social network sites (SNS), a battery of works has been conducted on online social networking. In the early days of the SNS market, studies were conducted on users' intention to accept [22–24]. Leng, Lada [23] shed light on SNS acceptance by employing intrinsic motivation, elements in the technology acceptance model (TAM), and entries in the theory of planned behavior (TPB). They validated that perceived usefulness significantly impacts attitude and behavior. It was also verified that perceived behavioral control and behavioral intention have a significant correlation. Moreover, perceived enjoyment was figured out to be the essential contributor to attitude. As the number of SNS users increases and the market grows, many studies have tried to explain the intention of continuous use [25–28]. Kim [29] explored the leading factors driving the continuance intention of SNS users. The author indicated that the major antecedents of continuance intention are usefulness, easiness, and interpersonal influence. After the general studies on continuance intention in academia were ripe, the word-of-mouth effect and recommendation intention started to be highlighted [30, 31]. Jo [30] identified antecedent factors of word-of-mouth of SNS users. The author confirmed that word of mouth is generated by continuance intention, usefulness, easiness, and pleasure. Lee, Kim [31] identified important variables that are significantly associated with the post-adoption behavior of SNS users. They introduced continuance intention and recommendation intention as post-assessment of users. The authors also reflected flow and user satisfaction as mediators. It was proved that emotion affects continuance intention through flow and user satisfaction. Afterward, users were able to try to switch to new services since latecomers appeared in the SNS market. Numerous studies have identified the factors that determine switching intention among SNSs [32–34]. Jo, Nam [34] elucidated the intention to switch between SNSs by expanding the TAM. They verified that easiness, alternative attractiveness, and peer influence are the preeminent drivers of switching intentions.

Along with the above studies, a number of researchers have performed the analysis of derivative functions and computational study of SNS. Kang, Kim [35] investigated the impact of social networks on the use of community-based knowledge services. They found that the centrality of the answering ties influences the quality of answers. It was demonstrated that centrality and the strength of co-answering ties hurt the quality of answers. In addition, several studies have quantitatively analyzed the behavior of social network users by using a tag, time, and usage frequency [36–38].

Some authors have investigated the use of social network and social media in the cases of academic performance. Al-Adwan, Albelbisi [6] built a conceptual framework for clarifying the precursors of academic performance among students. The authors asserted that easiness, usefulness, collaborative learning, enjoyment, and enhanced communication affect performance via social media use. Alamri, Almaiah [7] also examined the use of social media in the case of academic performance. It was demonstrated that easiness and usefulness influence performance through social media use. Sobaih, Hasanein [5] applied the TPB to educate the

formation mechanism of academic performance. They discovered that attitude, subjective norms, and perceived behavioral control impact performance via intention.

As the COVID-19 pandemic continues, the functions of social media and user behavior also changed. Vall-Roqué, Andrés [39] clarified the effects of social lockdown on SNS use, self-esteem, and body image disturbances among young women and adolescents. They revealed that the frequency of SNS usage increased to a statistically significant level during the social lockdown period. Furthermore, women have created appearance-focused Instagram accounts more frequently in this period. Pujadas-Hostench, Palau-Saumell [40] explored the intention to use the SNS brand page and its precursors to explain purchase intention on SNS. They unveiled that usage and gratification enhance both attitudes and intentions toward SNS brand pages. Qin, Kim [16] identified the leading factors affecting the behavioral intention of mobile SNS users. They analyzed the sample by dividing it into the United States and Korea. They found that enjoyment and subjective norms are the imperative contributors to behavioral intention. Zuo, Zhang [41] clarified the critical factors affecting social connectedness. They uncovered that sharing physical activity experiences, self-presentation, and positive comments are the main determinants of social connectedness. Chakraborty, Kumar [42] investigated the effects of adherence to social distancing and the psychological impact on SNS use. They discovered that respondents aged 21 to 35 were more likely to be active on social media when they were emotionally affected by COVID-19. Islam, Mäntymäki [43] explored the potential drawback of using SNS. They proved that COVID-19 fixation is brought on by both the danger of COVID-19 and unemployment. It was discovered that seeking emotional support through SNS cause SNS tiredness. Lee, Noh [44] looked into how Korean and Japanese residents felt about COVID-19. They examined the term frequency and corresponding shifts in interest in COVID-19 tweets from Korean and Japanese users. Four categories were used to group the words: issue, social distance, prevention, and emotion.

Putting the aforementioned research together, many studies have cast light on the behaviors of SNS users. Recent works gave a lens on behavioral changes of SNS users since the COVID-19 advent. However, there is insufficient research on how the risk factors of COVID-19 and the psychological pressure experienced by physical isolation affect SNS behavior.

## 2.2. Risk perception

Risk perception corresponds to an individual's cognitive understanding or emotional reaction to the likelihood that a certain event would do them harm [45]. Several studies have explored risk perception rather than actual risk since perceived risk is a pivotal antecedent to human behavior [46–48]. Previous studies have measured risk perception by dividing it into two scales [20, 49, 50]. Some scholars suggested an affective risk perception and a cognitive risk perception [18, 45, 50]. Other researchers proposed risk as feeling and risk analysis [20]. Affective risk perception and risk as feeling are similar in measuring indicators. Risk analysis and cognitive risk perception share similar measuring markers. This paper adopts the terms affective risk perception and cognitive risk perception to maintain consistency with the higher concept (i.e. risk perception).

Several studies have demonstrated that risk perception influences preventive behavior or social action during the pandemic [8, 18, 20]. Adiyoso and Wilopo [8] verified that risk perception is the critical determinant of subjective norms for social distancing. The authors showed that risk perception is the crucial predecessor of perceived behavioral control. Savadori and Lauriola [20] found that feelings of risk positively lead to participation in hygiene and social distancing. They also validated that risk analysis plays a key role in improving social distancing. Bae and Chang [18] explained the intention of citizens for tourism with contact to

hinder COVID-19 based on risk perception. They proved that affective risk perception significantly determines the behavioral intention of non-face-to-face tourism. Moreover, they showed that cognitive risk perception serves as the salient component in developing subjective norms and behavioral intentions.

In summary, a vast body of research has introduced risk perception to explicate preventive actions against common disasters or COVID-19 [8, 45, 51]. However, there was a dearth of works on the impacts of risk perception on subjective norms and perceived behavioral control on overall social measures, as well as social networking activities.

## 2.3. Cabin fever syndrome

Cabin fever refers to negative emotions such as discomfort and irritability experienced when an individual is kept alone in a quarantine location for an extended period [52–54]. This includes bad emotions and mental disorders such as helplessness, depression, and irritability caused by social restrictions [10]. Specifically, individuals experience a sense of isolation, loneliness, and restlessness due to the absence of social exchange [55]. Cabin fever syndrome is a common mental condition that anyone can suffer from long isolation [53].

The number of people who have experienced cabin fever syndrome has increased as individuals have less time to spend outside and interact with others due to the COVID-19 outbreak [56]. In this context, several studies have explored cabin fever syndrome. Estacio, Lumibao [10] investigated the cabin fever symptoms of university students and analyzed the differences between men and women. They described that many students experienced cabin fever even a little. The authors also reported that the female students had less focus and more appetite for food. Chakraborty, Kumar [42] examined the effect of several psychological influences, including cabin fever syndrome, on SNS use during the pandemic. They proved that psychological impact significantly increases social networking activities in people in their 2-30s. Chen, Bao [57] explored various symptoms of cabin fever caused by COVID-19. Manifestations include feelings of isolation, frustration, lethargy, emptiness, and delay. As Cabin fever syndrome was recognized as a new social problem, some studies suggested solutions to cope with [9, 11–13]. Solutions include following healthy living guidelines, being with nature, and connecting with others virtually.

As mentioned above, many studies investigated cabin fever syndrome, which is a psychological reaction experienced in isolation. However, little research attention has been paid to cabin fever syndrome on social networking intensity.

## 3. Research model and hypotheses

Fig 1 shows a conceptual model for identifying the main drivers of social networking intensity. A number of authors have revealed that risk perception in the COVID-19 environment affects citizens' reactions to social measures or protective behaviors [19, 58, 59]. Risk perception was found to drive social distancing attitude and social distancing intention [60]. People who think COVID-19 is more serious would recognize that people around them encourage social measures. As well, they may want to inject more resources into society's prevention policies. Thus, this work posits that affective risk perception and cognitive risk perception affect both subjective norms and perceived behavioral control. It was reported that cabin fever syndrome influences social distancing intention indirectly [54]. Moreover, cabin fever syndrome amplifies the SNS usage intensity [54]. Since social measures directly affect a sense of closure [61, 62], cabin fever syndrome would affect the responses of others and students to social measures. College students try to perform more social networking to compensate for reduced social activities and relieve emotional frustration [63, 64]. Therefore, this research explores the roles of cabin

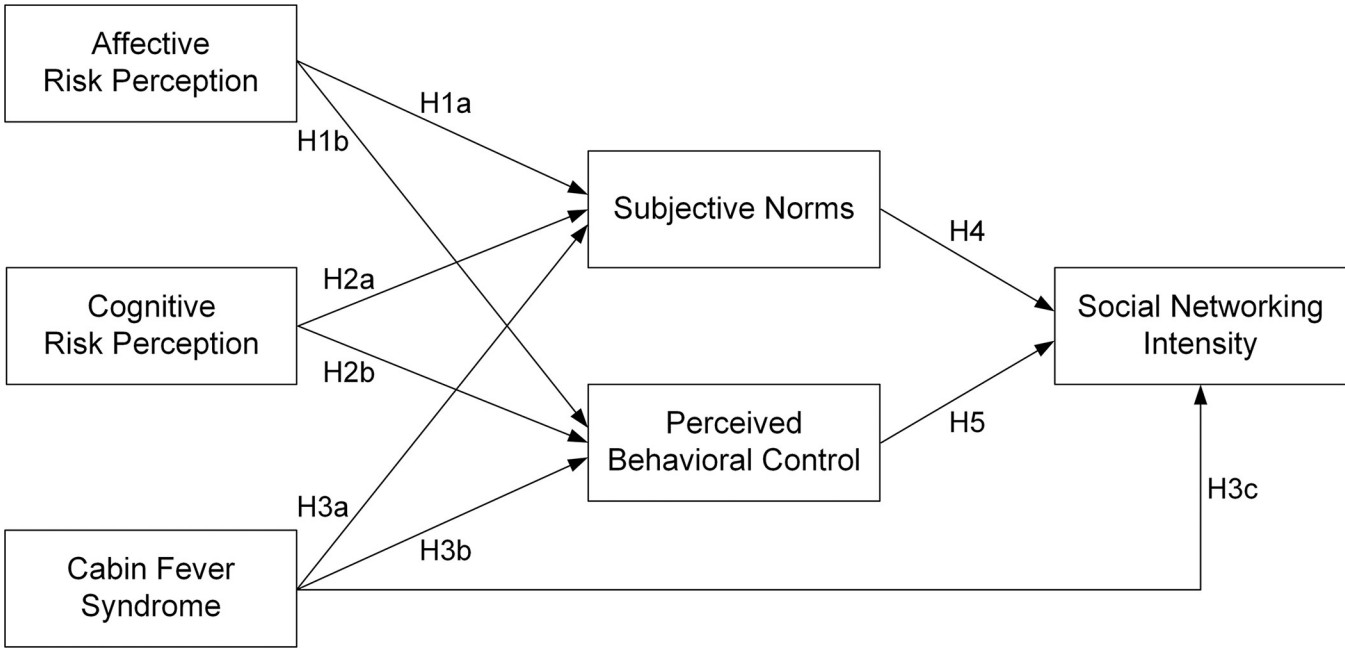

**Fig 1. Research model.**

fever syndrome in developing affective subjective norms, perceived behavioral control, and social networking intensity. Perceived behavioral control toward social measures promotes social distancing intention [51]. If students themselves have more resources devoted to social measures and people around them encourage to participate more in social measures, they will reduce outside activities. Ultimately, this increases social networking. Hence, the current study postulates that subjective norms and perceived behavioral control influence social networking intensity.

### 3.1. Affective risk perception

Affective risk perception is justified as the extent to which a person affectively responds against the threat of disaster [45]. Previous research empirically supported that risk perception influences both subjective norms and perceived behavioral control toward social distancing during COVID-19 [8]. Extended TPB with risk perception has proven significant in several risk contexts [65, 66]. The more emotionally university students perceive the dangers of COVID-19, the more they would try to follow social measures to prevent infection. They may adapt to the influences of their surroundings and control their behavior towards social measures. Given the above, affective risk perception is expected to improve subjective norms and perceived behavioral control.

H1a. Affective risk perception positively impacts subjective norms.
H1b. Affective risk perception positively impacts perceived behavioral control.

### 3.2. Cognitive risk perception

Cognitive risk perception is described as the extent to which a person objectively perceives the risk of disaster [45]. Cognitive risk perception is significantly associated with subjective norms [18]. University students encounter the daily updated COVID-19 situation through TV or online media. They recognize and understand the current risk of COVID-19 by looking at the number of confirmed cases, the number of deaths, and the vaccination rate. When confirmed

cases or dangerous events are reported, students may increase the level of subjective norms and perceived behavioral control on social measures to prevent risks. As a consequence, cognitive risk perception is believed to drive subjective norms and perceived behavioral control.

H2a. Cognitive risk perception positively impacts subjective norms.

H2b. Cognitive risk perception positively impacts perceived behavioral control.

### 3.3. Cabin fever syndrome

Cabin fever syndrome is justified as a negative mood with claustrophobic inertia when a person is trapped in a limited place for a long time [9, 42, 52]. One study found that the majority of university students experienced a none to mild cabin fever [10]. Risk factors originate from the many manifestations of cabin fever such as nihilism, procrastination, and obsessiveness [57]. University students with a higher cabin fever syndrome level will take social measures more because they bear the mental deprivation caused by physical isolation. They also would increase social networking to relieve psychological difficulties. Therefore, it is predicted that cabin fever syndrome leads to subjective norms, perceived behavioral control, and social networking intensity.

H3a. Cabin fever syndrome positively impacts subjective norms.

H3b. Cabin fever syndrome positively impacts perceived behavioral control.

H3c. Cabin fever syndrome positively impacts social networking intensity.

### 3.4. Subjective norms

Subjective norms measure the pressure placed on individuals by others who are important to him or her to act the behavior [67]. Previous research in information technology has found the significance of subjective norms on behavioral intention [15, 68, 69]. Subjective norms determine the intention to adopt and use in the context of SNS [16]. Among young people, subject norms play a pivotal role in shaping behavioral intention toward SNS [23, 70]. Empirical evidence in the prior study supports that subjective norms determine participation in SNS during the pandemic [71]. University students may form subjective norms by their friends and family. As the people around them emphasize social measures more, students will reduce outdoor activities and increase social media activities to interact with society. Based on this, this study suggests that subjective norms motivate social networking intensity.

H4. Subjective norms positively impact social networking intensity.

### 3.5. Perceived behavioral control

Perceived behavioral control is discussed as a person's cognition of his or her own ability to act on interest [72]. Past works have supported that perceived behavioral control facilitates the behavioral intention of SNS users [23, 40]. During COVID-19, perceived behavioral control significantly leads to an enhancement of active participation in SNS [71]. If university students are capable and willing to take social measures, they may be able to prevent COVID-19 by increasing their time at home. They will be more active in SNS activities to resolve psychological isolation, obtain information on COVID-19, and replace various physical activities. Judged from the above investigations, this paper predicts that perceived behavioral control amplifies social networking intensity.

H5. Perceived behavioral control positively impacts social networking intensity.

## 4. Empirical methodology

This study was approved by an institutional review board of RealSecu.

## 4.1. Data collection

The survey was carried out because it is considered the best method for analyzing the hypothesized relationships among the variables [73]. Ethical Committee of RealSecu approved this the survey based on the ethical criteria of research. We surveyed countries that are implementing social measures to prevent COVID-19 in accordance with the purpose of this study. At the time of the survey, South Korea was implementing social distancing, and Vietnam was in a state of social closure. An online questionnaire survey was delivered to university students in the two countries. We received responses from university students who had used social networking through their phones or PCs. The online questionnaire link was distributed between September 7[th] and 20[th] 2021. Some professors encouraged their undergraduate and postgraduate students to participate. Due to the unprecedented nature of the COVID-19 pandemic, the research used the snowball sampling technique to gather data. Snowball sampling is a useful method for identifying hard-to-reach populations or studying phenomena that are poorly understood [74]. After removing insincere and frivolous responses, 345 responses were analyzed. An a-priori sample size calculator was used to determine the minimum sample size required for structural equation models (SEM) [75]. By entering the necessary information, such as an anticipated effect size of 0.1, a desired statistical power level of 80%, 6 latent variables, 17 observed variables, and a probability level of 0.05, the minimum required sample size was determined to be 227. Since this study had a sample size of 345, this requirement was met. Among the final samples, 77 (21.8%) participants were Korean and 268 (77.8%) informants were Vietnamese. 128 students were male and 217 students were female. The mean age of the respondents was 21.084 with a standard deviation of 4.084. Table 1 shows the demographic features of the final sample.

## 4.2. Instrument

A questionnaire survey was performed to explore the university students' social networking intensity. The first page of the questionnaire included a description of informed consent. Informed consent was gathered in written form from all participants. The main body of questionnaire consisted of two parts. The first part asks for the demographic information of the participants. The second part comprises the major constructs such as risk perception, cabin fever syndrome, subjective norms, perceived behavioral control, and social networking intensity. All indicators were measured by a "7-point Likert scale". Since this work used a survey to test the research framework, all question items were selected from previous literature to verify the validity of the measurements. The measurement items were slightly adjusted to fit the SNS context. Before the main survey was performed, researchers in the related field thoroughly reviewed the questionnaire to confirm content validity. The responses were obtained from

**Table 1. Sample information.**

| Demographics | Item | Subjects (N = 345) | |
|---|---|---|---|
| | | Frequency | Percentage |
| Nation | South Korea | 77 | 22.3% |
| | Vietnam | 268 | 77.7% |
| Gender | Male | 128 | 37.1% |
| | Female | 217 | 62.9% |
| Age | 19 or younger | 48 | 13.9% |
| | 20–23 | 279 | 80.9% |
| | 24 or older | 18 | 5.2% |

11th June to 27th September 2021. The indicators of each construct are detailed in S1 Appendix.

## 5. Research results

### 5.1. Data analysis

In this study, SEM with partial least squares (PLS) was used to examine both the measurement model and the structural model. This work used SmartPLS 3.3.3 [76]. PLS is the most adequate technique for exploratory investigations [77]. Compared to covariance-based SEM, PLS has certain advantages in that there are fewer limitations on the distribution of sample size and residuals [78].

We analyzed the data with two main steps. In the first step, the measurement model was verified. We verified the reliability, validity, and discriminant validity of the scale. In the second step, the structural model was validated. We calculated the VIF value for multicollinearity verification. In addition, this research estimated the path coefficient, t-value, p-value, and $R^2$ values.

### 5.2. Measurement model

The present study checked construct reliability (Cronbach's alpha and composite reliability) and validity (convergent validity and discriminant validity) to confirm the measurement model. If Cronbach's alpha scores are over 0.7 and composite reliability (CR) estimates are above 0.70, reliability is ensured [79]. As detailed in Table 2, the scores of Cronbach's alpha ranged between 0.795 to 0.900 which exceeded the expected cut-off limit of 0.7 [80]. The CR scores are between 0.878 and 0.937, exceeding the acceptable threshold of 0.7. Hence, the construct reliability is deemed to show appropriate.

Convergent validity was evaluated by estimating the average variance extracted (AVE) and the factor loadings of each indicator. The lowest AVE (cabin fever syndrome) was 0.706, which is well over the recommended level of 0.5 [79]. The lowest factor loading (CFS2) was 0.825, supporting that the model has an adequate level of convergent validity [81].

**Table 2. Scale reliability and validity.**

| Construct | Items | Mean | St. Dev. | Factor Loading | Cronbach's Alpha | CR | AVE |
|---|---|---|---|---|---|---|---|
| Affective Risk Perception | ARP1 | 5.217 | 1.611 | 0.882 | 0.872 | 0.921 | 0.794 |
| | ARP2 | 5.545 | 1.611 | 0.904 | | | |
| | ARP3 | 5.652 | 1.455 | 0.888 | | | |
| Cognitive Risk Perception | CRP1 | 5.539 | 1.414 | 0.926 | 0.802 | 0.909 | 0.834 |
| | CRP2 | 5.174 | 1.459 | 0.900 | | | |
| Cabin Fever Syndrome | CFS1 | 4.661 | 1.304 | 0.854 | 0.795 | 0.878 | 0.706 |
| | CFS2 | 5.191 | 1.331 | 0.825 | | | |
| | CFS3 | 4.901 | 1.318 | 0.841 | | | |
| Subjective Norms | SNO1 | 5.838 | 1.583 | 0.831 | 0.900 | 0.937 | 0.834 |
| | SNO2 | 6.104 | 1.283 | 0.954 | | | |
| | SNO3 | 6.116 | 1.333 | 0.948 | | | |
| Perceived Behavioral Control | PBC1 | 5.994 | 1.441 | 0.938 | 0.891 | 0.933 | 0.822 |
| | PBC2 | 5.988 | 1.333 | 0.923 | | | |
| | PBC3 | 5.817 | 1.642 | 0.857 | | | |
| Social Networking Intensity | SNI1 | 5.542 | 1.343 | 0.935 | 0.858 | 0.915 | 0.783 |
| | SNI2 | 5.545 | 1.476 | 0.936 | | | |
| | SNI3 | 5.186 | 1.437 | 0.775 | | | |

**Table 3. Correlation matrix and discriminant evaluation.**

| Construct | 1 | 2 | 3 | 4 | 5 | 6 |
|---|---|---|---|---|---|---|
| 1. Affective Risk Perception | 0.892 | | | | | |
| 2. Cognitive Risk Perception | 0.481 | 0.909 | | | | |
| 3. Cabin Fever Syndrome | 0.174 | 0.386 | 0.840 | | | |
| 4. Subjective Norms | 0.339 | 0.336 | 0.122 | 0.913 | | |
| 5. Perceived Behavioral Control | 0.400 | 0.339 | 0.195 | 0.709 | 0.907 | |
| 6. Social Network Intensity | 0.247 | 0.275 | 0.549 | 0.345 | 0.435 | 0.885 |

Note: Diagonal elements are the square root of AVE

Discriminant validity was confirmed by [79] criterion and HTMT [82]. The criterion for discriminant validity, as recommended by [79], was satisfied. The square root of AVE was greater than the inter-variable correlation coefficients, as shown in Table 3.

Furthermore, discriminant validity was established by checking the HTMT (Heterotrait-Monotrait Ratio of Correlations) values, with all constructs having HTMT values below the recommended threshold of 0.85, as shown in Table 4 [83].

## 5.3. Structural model

This research estimates the coefficient of determination ($R^2$) and path coefficients through a bootstrapping resampling method (5000 re-samples) [84]. Fig 2 displays $R^2$ and the coefficients for each path. Six of the nine hypotheses in the research model are supported.

The use of SEM did not reveal any issues with multicollinearity, as indicated by the VIF values, which were all below 5. Specifically, the VIF values for the constructs were as follows: Affective risk perception = 1.301, Cognitive risk perception = 1.483, Cabin fever syndrome = 1.040, Subjective norms = 2.013, and Perceived behavioral control = 2.062.

Consistent with hypotheses, affective risk perception is significantly related to both subjective norms and perceived behavioral control. Therefore, H1a and H1b are accepted. In line with expectations, cognitive risk perception significantly influences both subjective norms and perceived behavioral control. Hence, H2a and H2b are supported. Cabin fever syndrome does not impact both subjective norms and perceived behavioral control, failing to adopt H3a and H3b. However, it positively affects social networking intensity, providing empirical support for H3c. Contrary to expectations, subjective norms do not influence social networking intensity. Hence, H4 is not supported. As predicted, perceived behavioral control is significantly related to social networking intensity. Thus, H5 is supported. The structural model describes 41.8% of the variance in social networking intensity, 15.4% of the variance in subjective norms, and 19.3% of the variance in perceived behavioral control. Table 5 summarizes the analysis results.

**Table 4. HTMT.**

| Construct | 1 | 2 | 3 | 4 | 5 | 6 |
|---|---|---|---|---|---|---|
| 1. Affective Risk Perception | | | | | | |
| 2. Cognitive Risk Perception | 0.559 | | | | | |
| 3. Cabin Fever Syndrome | 0.218 | 0.522 | | | | |
| 4. Subjective Norms | 0.366 | 0.366 | 0.130 | | | |
| 5. Perceived Behavioral Control | 0.446 | 0.380 | 0.219 | 0.785 | | |
| 6. Social Network Intensity | 0.276 | 0.324 | 0.635 | 0.394 | 0.497 | |

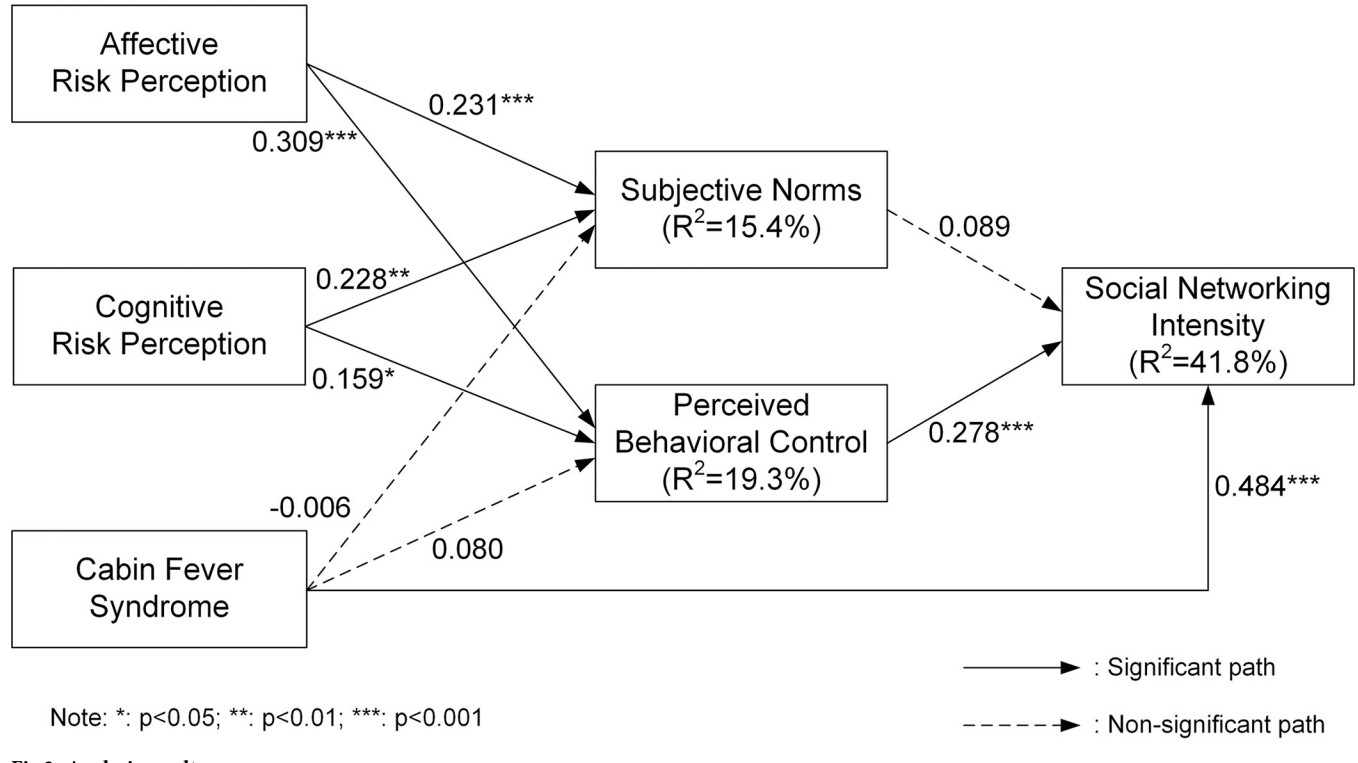

Note: *: p<0.05; **: p<0.01; ***: p<0.001

**Fig 2. Analysis results.**

## 6. Discussion

This paper aimed to identify factors impacting the social networking intensity among university students during isolation. This has been achieved by reflecting risk perception, the sense of isolation, subjective norm, and perceived behavioral control in the research model.

The analysis results demonstrated that affective risk perception and cognitive risk perception are significantly associated with the subjective norms. The same results were obtained in a past study, in which risk perception influences subjective norms for social distancing [8]. These results could be explained by the reason that when university students perceive that COVID-19 is more dangerous, their acquaintances also perceive it to be riskier and may emphasize social measures more.

**Table 5. Summary of the results.**

| H | Cause | Effect | Coefficient | T-value | Hypothesis |
|---|---|---|---|---|---|
| H1a | Affective Risk Perception | Subjective Norms | 0.231 | 3.794 | Supported |
| H1b | Affective Risk Perception | Perceived Behavioral Control | 0.309 | 4.929 | Supported |
| H2a | Cognitive Risk Perception | Subjective Norms | 0.228 | 3.083 | Supported |
| H2b | Cognitive Risk Perception | Perceived Behavioral Control | 0.159 | 2.187 | Supported |
| H3a | Cabin Fever Syndrome | Subjective Norms | -0.006 | 0.097 | Not Supported |
| H3b | Cabin Fever Syndrome | Perceived Behavioral Control | 0.080 | 1.450 | Not Supported |
| H3c | Cabin Fever Syndrome | Social Networking Intensity | 0.484 | 10.430 | Supported |
| H4 | Subjective Norms | Social Networking Intensity | 0.089 | 1.521 | Not Supported |
| H5 | Perceived Behavioral Control | Social Networking Intensity | 0.278 | 4.198 | Supported |

In keeping with the results derived from a prior study [8], the findings revealed that affective risk perception and cognitive risk perception are the major deciding factors of perceived behavioral control. [54] reported that affective risk perception influences social distancing attitude and cognitive risk perception influences social distancing intention. As well, [51] argued that risk perception impacts attitude towards e-learning via social distancing attitude and social distancing intention. These significant relationships could be elucidated by the fact that university students are more likely to comply with social measures when they feel more at risk for COVID-19.

The analysis corroborated that cabin fever syndrome does not influence subjective norms and perceived behavioral control. We can make the following inference. Groups that feel more closed may want to temporarily do more outside activities to relieve their frustration. At the same time, they may ultimately seek to comply more with social measures to end the pandemic earlier. In other words, people seem to have a desire to violate social measures to eliminate frustration in the short term and to comply with social measures to end transmission in the long term. This offset each other's effects, leading to insignificant effects of cabin fever syndrome on subjective norms and perceived behavioral control. The empirical findings provided evidence that cabin fever syndrome significantly facilitates social networking intensity. This observation is congruent with the conclusion in [42, 54]. These results lie in the fact that students with stronger cabin fever syndrome use SNS more. They may have experienced psychological problems in addition to the risks of COVID-19. Students with a stronger feeling of social isolation would resolve mental pressures by communicating with others in cyberspace.

The results indicate that subjective norms are insignificant factors in shaping social networking intensity. A possible explanation for this result is that the actual frequency of use does not increase even if the people around the university students emphasize social measures. Students may respond to COVID-19 based on their judgment rather than the influence of those around them. The findings show that perceived behavioral control has a significant positive effect on social networking intensity. Similar to this result, empirical studies support that control has a significant effect on social distancing intention [8] and social distancing behavior [17]. The rationale for these results would be that social networking activities do not correspond to social measures, but they are windows to interact with disconnected society and find psychological stability during an isolated period. Students who have sufficient personal conditions to comply with social measures will refrain from going out and, consequently, increase social media activity.

## 7. Theoretical and practical implications

### 7.1. Theoretical contributions

This paper draws several valuable theoretical contributions to academia. First, the current work provides a novel contribution to researchers in that it proposes a model by modifying TPB and combining risk factors and psychological components to predict the social networking intensity of university students during the pandemic. This was a very meaningful attempt because former research has mainly paid attention to communicative values and technical elements to explain SNS behavior. Second, this work proved the significant influences of risk perception on the subjective norm and perceived behavioral control. Extant literature suggested belief, perceived usefulness, or perceived ease of use as antecedent factors of subjective norms [8, 15, 85]. Some scholars have proposed belief, internet self-efficacy, or self-esteem as leading factors of perceived behavioral control [85, 86]. This work introduced a new variable specific to COVID-19 and clarified their role in the process of forming subjective norms and perceived behavioral control. Third, this study explained SNS behavior by introducing uncomfortable

feelings that college students feel due to isolation. Through this, it is inferred that students with a higher sense of social isolation may increase social networking activities to alleviate psychological frustration. Lastly, the present study makes a contribution in that it explicated social networking intensity based on subjective norms and perceived behavioral control for social measures, not social networking itself. Empirical analyzes have demonstrated that human behavior in response to external events (i.e. social measures) can cause other associated actions (i.e. social networking) in special circumstances such as pandemic).

## 7.2. Practical implications

This paper derives many practical implications for practitioners. First, the study proved that risk perception drives both subjective norms and perceived behavioral control. Thus, SNS providers should guide university students to participate in social measures by performing a function of public interest. The government should examine university students' perceptions of the dangers of COVID-19 and emphasize the seriousness of the contagion to a group with a relatively low level of risk awareness. Second, the analysis validated that cabin fever syndrome amplifies online social networking. Hence, providers should pay special attention to the group with a high degree of cabin fever syndrome among university students. They will have to create a new platform that can reduce the feeling of isolation or encourage more realistic social exchange through SNS. Governments must apply somewhat relaxed distancing or implement living health guidelines for constructive social networking. Last, the study showed that the higher the perceived behavioral control over social measures, the higher the frequency of SNS use. Marketers need to undertake a questionnaire on social measures for university students to identify their level of perceived behavioral control. Providers can perform specialized marketing or events according to the level. For example, managers can show the cases of COVID-19 risk more frequently to a group with a lower perception of behavioral control to inform the COVID-19 more realistic. This eventually encourages people to adhere to social measures more.

## 8. Conclusion and further research

COVID-19 has terrified people around the world and has taken a social, economic, and cultural hit. Citizens have practiced social distancing and university students have taken classes through online lectures. This study identified factors influencing social networking intensity in the COVID-19 era. Constructs contain risk perception, cabin fever syndrome, and the components of TPB. To confirm the explanatory strength of the developed model, the data collected through the questionnaire were analyzed with the SEM. The study results showed that two types of risk perception lead to subjective norms and perceived behavioral control. The findings demonstrated that cabin fever syndrome stimulates social networking. The analysis revealed that perceived behavioral control triggers social networking intensity.

Although this paper proposed and tested new hypotheses, there are still issues that need to be resolved in the follow-up study. First, this research investigated only university students' social networking. These days, SNS has commercial and educational purposes, not just for daily sharing or communication with acquaintances. Therefore, future research should survey various age groups to enhance the validity of this study. Second, the present work introduced only risk and mental exhaustion among various exogenous variables that could be considered due to COVID-19. In addition, the driving force of the government and policies, sanitary reinforcement, vaccination, and the declaration of 'With Corona' by advanced countries may also affect SNS activities. In a follow-up study, it would be of academic value to introduce variables more comprehensively. Third, the results of this study may appear differently depending on

the development of the COVID-19 situation. Risk perception may be lowered by countries with higher vaccination rates and improved hygiene awareness among citizens. Therefore, future studies are necessary to derive new findings by re-verifying the research model with a gap of several months.

## Supporting information

**S1 Appendix.**
(DOCX)

**S1 Data.**
(ZIP)

**S1 File.**
(CSV)

## Author Contributions

**Conceptualization:** Hyeon Jo.

**Data curation:** Hyeon Jo.

**Formal analysis:** Hyeon Jo.

**Investigation:** Hyeon Jo.

**Methodology:** Hyeon Jo.

**Writing – original draft:** Hyeon Jo.

**Writing – review & editing:** Eun-Mi Baek.

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
