## [Decision Letter · Decision Letter 0]

15 Feb 2023

PONE-D-23-02960Impacts of Social Isolation and Risk Perception on Social Networking Intensity Among University Students During COVID-19PLOS ONE

Dear Dr. Jo,

Thank you for submitting your manuscript to PLOS ONE. After careful consideration, we feel that it has merit but does not fully meet PLOS ONE’s publication criteria as it currently stands. Therefore, we invite you to submit a revised version of the manuscript that addresses the points raised during the review process.

We look forward to receiving your revised manuscript.

Kind regards,

Ahmad Samed Al-Adwan

Academic Editor

PLOS ONE

Journal Requirements: Please follow the instructions and template of the Journal. 

Additional Editor Comments:

Both reviewers believe that your research paper has a merit. However, there are major concerns that need to be addressed. The reviewers' comments are highlighted below. Please make sure to address all these comments. 

Reviewers' comments:

Reviewer's Responses to Questions

**Comments to the Author**

1. Is the manuscript technically sound, and do the data support the conclusions?

Reviewer #1: Partly

Reviewer #2: Yes

2. Has the statistical analysis been performed appropriately and rigorously? 

Reviewer #1: Yes

Reviewer #2: Yes

3. Have the authors made all data underlying the findings in their manuscript fully available?

Reviewer #1: Yes

Reviewer #2: Yes

4. Is the manuscript presented in an intelligible fashion and written in standard English?

Reviewer #1: Yes

Reviewer #2: No

5. Review Comments to the Author

Reviewer #1: 1. The novelty of this paper is not clear. The authors need to clearly highlight the research problems, significance, and gaps in the literature. Furthermore, it would be very useful to include a clear research objective(s) or question(s). such novelty should be also reflected in the abstract section.

2. The authors need to include a paragraph at the end of the introduction section to highlight the structure of the paper.

3. The methodology section is fine. However, additional information can be useful. The authors are advised to provide information about when data were collected, the adequacy of snow sampling approach for study, and the appropriateness of the sample size obtained.

4. while data analysis is well-reported, the authors need to report the multi-collinearity test. In addition, it is important to provide a small paragraph at the beginning of the analysis section to highlight the main steps followed for data analysis (e.g., measurement model and structural model).

5. the discussion section can be enhanced by comparing the findings of this study with those reported in the previous ones. Further discussion is required for the hypotheses that were found not supported.

6. The introduction and literature review should be improved by including research about social media adoption by students. This includes:

· Investigating the Impact of Social Media Use on Student’s Perception of Academic Performance in Higher Education: Evidence from Jordan

· Social media applications affecting Students' academic performance: A model developed for sustainability in higher education

· Higher education in and after COVID-19: The impact of using social network applications for e-learning on students' academic performance.

Reviewer #2: -The significance of the study should be introduced in a better. Additionally, the research objectives or questions should be clearly highlighted.

-The theoretical foundation should be extensively explained. This should be done in section 3.

- The methodology section needs major modifications and more details. This includes: the procedure followed to collect data, clear identification of the study participants, appropriateness of sampling method, and sample size.

- The author needs to report the HTMT test for discriminant validity.

6. PLOS authors have the option to publish the peer review history of their article (what does this mean?). If published, this will include your full peer review and any attached files.

Reviewer #1: No

Reviewer #2: No

---

## [Author Response · Author response to Decision Letter 0]

8 Mar 2023

Impacts of Social Isolation and Risk Perception on Social Networking Intensity Among University Students During COVID-19

1st Response Letter

Editors in Chief

Dear Ahmad Samed Al-Adwan, 

Thank you for reviewing the paper. Also, thank you very much for letting us know the result of major revision. We did our best to correct the issues pointed out by reviewers. We have answered and corrected the points you pointed out for each item below. We saved the manuscript 1) in two separate ways (Revised Manuscript with Track Changes and Manuscript) as the editor guided us. Thank you for reviewing the paper. 

Reviewer(s)' Comments to Author:

Reviewer: 1

1. The novelty of this paper is not clear. The authors need to clearly highlight the research problems, significance, and gaps in the literature. Furthermore, it would be very useful to include a clear research objective(s) or question(s). such novelty should be also reflected in the abstract section.

Dear Reviewer 1, 

Thank you for judging our paper. Thank you for pointing out the research problems, importance, research gap, research purpose, and novelty of our research. We supplemented the revised version with contents that can highlight novelty as follows. Thank you.

1 Location Page 5, 3rd paragraph, 1st line. (Location is based on the revised manuscript.)

 From ㅡ

 To Risk perception, social measures (e.g. social distancing), and feelings of closure have a causal relationship. The spread of COVID-19 has made people aware of risk. To prevent infection, humans have implemented various social measures. Social measures have increased feelings of claustrophobia by curbing outdoor activities and increasing time spent at home. Risk perception, social measures, and feelings of closure may organically influence human behaviors. However, the problem with existing studies is that they introduce these three variables independently [18, 19] or selectively include two measures [20, 21]. This study is of academic significance because it addresses this research gap. In this sense, we posit the following research questions. 

1. Do people with higher levels of risk perception seek to comply with social measures more? 

2. Do people who feel more enclosed try to comply with social measures more? 

3. Does closedness affect social networking behavior directly? 

The objectives of this study are to 1) examine how risk of COVID-19 and personal dispositions toward lockdown influence responses to social measures, 2) identify the impact of individual responses to social measures on social networking activity, and 3) test the direct impact of feelings of lockdown on social networking.

2. The authors need to include a paragraph at the end of the introduction section to highlight the structure of the paper.

Thank you for your good opinion on the overall composition of our paper. As suggested by the reviewer, the contents were supplemented as follows.

1 Location Page 6, 4th paragraph, 1st line.

 From ㅡ

 To The paper is organized as follows. Section 2 lists prior research related to social networking, risk perception, and cabin fever syndrome. Section 3 describes the research model and hypotheses. Section 4 presents the data collection and measurement tools for the empirical analysis. Section 5 presents the results of the study. Section 6 contains a discussion of the results. Section 7 presents theoretical and practical implications. Finally, Section 8 discusses the limitations of the study and future research directions.

3. The methodology section is fine. However, additional information can be useful. The authors are advised to provide information about when data were collected, the adequacy of snow sampling approach for study, and the appropriateness of the sample size obtained.

Thank you for your valuable advice on the process of collecting data on our paper. We supplemented the revised version with the timing of data collection, the basis for applying snow sampling, and the academic thinking of the minimum sample size as follows. Thank you.

1 Location Page 20, 2nd paragraph, 8th line.

 From ㅡ

 To The online questionnaire link was distributed between September 7th and 20th 2021. Some professors encouraged their undergraduate and postgraduate students to participate. Due to the unprecedented nature of the COVID-19 pandemic, the research used the snowball sampling technique to gather data. Snowball sampling is a useful method for identifying hard-to-reach populations or studying phenomena that are poorly understood [74]. After removing insincere and frivolous responses, 345 responses were analyzed. An a-priori sample size calculator was used to determine the minimum sample size required for structural equation models (SEM) [75]. By entering the necessary information, such as an anticipated effect size of 0.1, a desired statistical power level of 80%, 6 latent variables, 17 observed variables, and a probability level of 0.05, the minimum required sample size was determined to be 227. Since this study had a sample size of 345, this requirement was met.

4. while data analysis is well-reported, the authors need to report the multi-collinearity test. In addition, it is important to provide a small paragraph at the beginning of the analysis section to highlight the main steps followed for data analysis (e.g., measurement model and structural model).

Thank you for pointing out the multicollinearity of our research data and the analysis procedure. We tested multicollinearity based on the VIF value as below and supplemented the overall analysis procedure.

1 Location Page 23, 4th paragraph, 1st line.

 From ㅡ

 To We analyzed the data with two main steps. In the first step, the measurement model was verified. We verified the reliability, validity, and discriminant validity of the scale. In the second step, the structural model was validated. We calculated the VIF value for multicollinearity verification. In addition, this research estimated the path coefficient, t-value, p-value, and R2 values.

2 Location Page 27, 2nd paragraph, 1st line.

 From ㅡ

 To The use of SEM did not reveal any issues with multicollinearity, as indicated by the VIF values, which were all below 5. Specifically, the VIF values for the constructs were as follows: Affective risk perception = 1.301, Cognitive risk perception = 1.483, Cabin fever syndrome = 1.040, Subjective norms = 2.013, and Perceived behavioral control = 2.062.

5. the discussion section can be enhanced by comparing the findings of this study with those reported in the previousones. Further discussion is required for the hypotheses that were found not supported.

Thank you for your valuable advice on the comparison of our paper's discussion with previous studies and the explanation of the rejected hypothesis. We agree with the reviewer. The corresponding part was modified as follows.

1 Location Page 29, 3rd paragraph, 2nd line.

 From ㅡ

 To The analysis results demonstrated that affective risk perception and cognitive risk perception are significantly associated with the subjective norms. The same results were obtained in a past study, in which risk perception influences subjective norms for social distancing [8]. Thess results could be explained by the reason that when university students perceive that COVID-19 is more dangerous, their acquaintances also perceive it to be riskier and may emphasize social measures more.

In keeping with the results derived from a prior study [8], the findings revealed that affective risk perception and cognitive risk perception are the major deciding factors of perceived behavioral control. [54] reported that affective risk perception influences social distancing attitude and cognitive risk perception influences social distancing intention. As well, [51] argued that risk perception impacts attitude towards e-learning via social distancing attitude and social distancing intention. These significant relationships could be elucidated by the fact that university students are more likely to comply with social measures when they feel more at risk for COVID-19.

Tha analysis corroborated that cabin fever syndrome does not influence subjective norms and perceived behavioral control. We can make the following inference. Groups that feel more closed may want to temporarily do more outside activities to relieve their frustration. At the same time, they may ultimately seek to comply more with social measures to end the pandemic earlier. In other words, people seem to have a desire to violate social measures to eliminate frustration in the short term and to comply with social measures to end transmission in the long term. This offset each other's effects, leading to insignificant effects of cabin fever syndrome on subjective norms and perceived behavioral control. The empirical findings provided evidence that cabin fever syndrome significantly facilitates social networking intensity. This observation is congruent with the conclusion in [42, 54]. These results lie in the fact that students with stronger cabin fever syndrome use SNS more.

6. The introduction and literature review should be improved by including research about social media adoption by students. This includes:

· Investigating the Impact of Social Media Use on Student’s Perception of Academic Performance in HigherEducation: Evidence from Jordan

· Social media applications affecting Students' academic performance: A model developed for sustainability in higher education

· Higher education in and after COVID-19: The impact of using social network applications for e-learning on students' academic performance.

Thank you for presenting a good prior study related to our paper. We reflected the study in the introduction and theoretical background as follows. Thank you very much for reviewing our paper.

1 Location Page 3, 1st paragraph, 6th line.

 From ㅡ

 To Sometimes, students use social netowrk apps [5] or social media [6, 7] to enhance academical performance.

2 Location Page 8, 3rd paragraph, 1st line.

 From ㅡ

 To Some authors have investigated the use of social network and social media in the cases of academic performance. Al-Adwan, Albelbisi (6) built a conceptual framework for clarifying the precursors of academic performance among students. The authors asserted that easiness, usefulness, collaborative learning, enjoyment, and enhanced communication affect performance via social media use. Alamri, Almaiah (7) also examined the use of social media in the case of academic performance. It was demonstrated that easiness and usefulness influence performance through social media use. Sobaih, Hasanein (5) applied the TPB to eludicate the formation mechanism of academic performance. They discovered that attitude, subjective norms, and perceived behavioral condtrol impact performance via intention.

Reviewer: 2

- The significance of the study should be introduced in a better. Additionally, the research objectives orquestions should be clearly highlighted.

Dear Reviewer 2, 

Thank you for judging our paper. Thank you for pointing out the significance and research obejctives of our research. We supplemented the revised version with contents that can highlight importance and purposes as follows. Thank you.

1 Location Page 5, 3rd paragraph, 1st line. (Location is based on the revised manuscript.)

 From ㅡ

 To Risk perception, social measures (e.g. social distancing), and feelings of closure have a causal relationship. The spread of COVID-19 has made people aware of risk. To prevent infection, humans have implemented various social measures. Social measures have increased feelings of claustrophobia by curbing outdoor activities and increasing time spent at home. Risk perception, social measures, and feelings of closure may organically influence human behaviors. However, the problem with existing studies is that they introduce these three variables independently [18, 19] or selectively include two measures [20, 21]. This study is of academic significance because it addresses this research gap. In this sense, we posit the following research questions. 

1. Do people with higher levels of risk perception seek to comply with social measures more? 

2. Do people who feel more enclosed try to comply with social measures more? 

3. Does closedness affect social networking behavior directly? 

The objectives of this study are to 1) examine how risk of COVID-19 and personal dispositions toward lockdown influence responses to social measures, 2) identify the impact of individual responses to social measures on social networking activity, and 3) test the direct impact of feelings of lockdown on social networking.

- The theoretical foundation should be extensively explained. This should be done in section 3.

Thank you for pointing out the theoretical foundation of our research. We agree with the reviewer. We further read the preceding studies as follows to reinforce the theoretical foundation in Section 3.

1 Location Page 14, 2nd paragraph, 1st line.

 From Figure 1 shows a conceptual model for identifying the main drivers of social networking intensity. This work posits that affective risk perception and cognitive risk perception affect both subjective norms and perceived behavioral control. It explores the roles of cabin fever syndrome in developing affective subjective norms, perceived behavioral control, and social networking intensity. The current study postulates that subjective norms and perceived behavioral control influence social networking intensity. 

 To Fig 1 shows a conceptual model for identifying the main drivers of social networking intensity. A number of authors have revealed that risk perception in the COVID-19 environment affects citizens' reactions to social measures or protective behaviors [19, 58, 59]. Risk perception was found to drive social distancing attitude and social distancing intention [60]. People who think COVID-19 is more serious would recognize that people around them encourage social measures. As well, they may want to inject more resources into society's prevention policies. Thus, this work posits that affective risk perception and cognitive risk perception affect both subjective norms and perceived behavioral control. It was reported that cabin fever syndrome influences social distancing intention indirectly [54]. Morever, cabin fever synrome amplifies ths SNS usage intensity [54]. Since social measures directly affect a sense of closure [61, 62], cabin fever syndrome would affect the responses of others and students to social measures. College students try to perform more social networking to compensate for reduced social activities and relieve emotional frustration [63, 64]. Therefore, this research explores the roles of cabin fever syndrome in developing affective subjective norms, perceived behavioral control, and social networking intensity. Perceived behavioral control toward social measures promotes social distancing intention [51]. If students themselves have more resources devoted to social measures and people around them encourage to participate more in social measures, they will reduce outside activities. Ultimately, this increases social networking. Hence, the current study postulates that subjective norms and perceived behavioral control influence social networking intensity. 

- The methodology section needs major modifications and more details. This includes: the procedure followed to collect data, clear identification of the study participants, appropriateness of sampling method, and sample size.

Thank you for your valuable advice on the process of collecting data on our paper. We supplemented the revised version with the timing of data collection, the basis for applying snow sampling, and the academic thinking of the minimum sample size as follows. Thank you.

1 Location Page 20, 2nd paragraph, 3th line.

 From ㅡ

 To We surveyed countries that are implementing social measures to prevent COVID-19 in accordance with the purpose of this study. At the time of the survey, South Korea was implementing social distancing, and Vietnam was in a state of social closure. An online questionnaire survey was delivered to university students in the two countries. We received responses from university students who had used social networking through their phones or PCs. The online questionnaire link was distributed between September 7th and 20th 2021. Some professors encouraged their undergraduate and postgraduate students to participate. Due to the unprecedented nature of the COVID-19 pandemic, the research used the snowball sampling technique to gather data. Snowball sampling is a useful method for identifying hard-to-reach populations or studying phenomena that are poorly understood [74]. After removing insincere and frivolous responses, 345 responses were analyzed. An a-priori sample size calculator was used to determine the minimum sample size required for structural equation models (SEM) [75]. By entering the necessary information, such as an anticipated effect size of 0.1, a desired statistical power level of 80%, 6 latent variables, 17 observed variables, and a probability level of 0.05, the minimum required sample size was determined to be 227.

- The author needs to report the HTMT test for discriminant validity.

Thank you for enhancing the rigor of verifying the discriminant validity of our research model. We presented HTMT as follows. Thank you very much for reviewing our research and giving us valuable advice.

Page 26, 1st paragraph, 1th line.

Furthermore, discriminant validity was established by checking the HTMT (Heterotrait-Monotrait Ratio of Correlations) values, with all constructs having HTMT values below the recommended threshold of 0.85, as shown in Table 4 [83].

Table 4. HTMT

Construct 1 2 3 4 5 6

1. Affective Risk Perception 

2. Cognitive Risk Perception 0.559 

3. Cabin Fever Syndrome 0.218 0.522 

4. Subjective Norms 0.366 0.366 0.130 

5. Perceived Behavioral Control 0.446 0.380 0.219 0.785 

6. Social Network Intensity 0.276 0.324 0.635 0.394 0.497 



---

## [Editor Report · Decision Letter 1]

21 Mar 2023

Impacts of Social Isolation and Risk Perception on Social Networking Intensity Among University Students During COVID-19

PONE-D-23-02960R1

Dear Dr. Baek,

We’re pleased to inform you that your manuscript has been judged scientifically suitable for publication and will be formally accepted for publication once it meets all outstanding technical requirements.

Kind regards,

Ahmad Samed Al-Adwan

Academic Editor

PLOS ONE

Additional Editor Comments (optional):

Thank you for resubmitting the revised version of your paper. The paper has improved significantly after addressing the reviewers' comment.
---

## [Editor Report · Acceptance letter]

20 Apr 2023

PONE-D-23-02960R1 

Impacts of Social Isolation and Risk Perception on Social Networking Intensity Among University Students During COVID-19 

Dear Dr. Baek:

I'm pleased to inform you that your manuscript has been deemed suitable for publication in PLOS ONE. Congratulations! Your manuscript is now with our production department. 

Kind regards, 

on behalf of

Prof. Ahmad Samed Al-Adwan 

Academic Editor

PLOS ONE